# MRI Radiomics and Machine Learning for the Prediction of Oncotype Dx Recurrence Score in Invasive Breast Cancer

**DOI:** 10.3390/cancers15061840

**Published:** 2023-03-18

**Authors:** Valeria Romeo, Renato Cuocolo, Luca Sanduzzi, Vincenzo Carpentiero, Martina Caruso, Beatrice Lama, Dimitri Garifalos, Arnaldo Stanzione, Simone Maurea, Arturo Brunetti

**Affiliations:** 1Department of Advanced Biomedical Sciences, University of Naples “Federico II”, 80131 Naples, Italy; 2Department of Medicine, Surgery, and Dentistry, University of Salerno, 84084 Baronissi, Italy; 3Augmented Reality for Health Monitoring Laboratory (ARHeMLab), Department of Electrical Engineering and Information Technology, University of Naples “Federico II”, 80131 Naples, Italy

**Keywords:** radiomics, oncotype DX, breast cancer, machine learning, neoadjuvant chemotherapy, magnetic resonance, recurrence score

## Abstract

**Simple Summary:**

The preoperative definition of the Oncotype DX recurrence score would be critical to identify breast cancer patients who could benefit from chemotherapy before surgery. In this work, we built a machine learning model applied to DCE-MRI images of a publicly available dataset to predict the Oncotype DX score in patients with breast cancer. As a result, the model achieved an accuracy of 60% in the training set and 63% (AUC = 0.66) in the test set. Our findings support the feasibility of radiomics and machine learning for the prediction of prognostic data in breast cancer, encouraging further, preferably multicenter, investigations to further improve the performance of the model and assess its generalizability.

**Abstract:**

Aim: To non-invasively predict Oncotype DX recurrence scores (ODXRS) in patients with *ER*+ *HER2*- invasive breast cancer (IBC) using dynamic contrast-enhanced (DCE) MRI-derived radiomics features extracted from primary tumor lesions and a ML algorithm. Materials and Methods: Pre-operative DCE-MRI of patients with IBC, no history of neoadjuvant therapy prior to MRI, and for which the ODXRS was available, were retrospectively selected from a public dataset. ODXRS was obtained on histological tumor samples and considered as positive if greater than 16 and 26 in patients aged under and over 50 years, respectively. Tumor lesions were manually annotated by three independent operators on DCE-MRI images through 3D ROIs positioning. Radiomic features were therefore extracted and selected using multistep feature selection process. A logistic regression ML classifier was then employed for the prediction of ODXRS. Results: 248 patients were included, of which 87 with positive ODXRS. 166 (66%) patients were grouped in the training set, while 82 (33%) in the test set. A total of 1288 features was extracted. Of these, 1244 were excluded as 771, 82 and 391 were excluded as not stable (*n* = 771), not variant (*n* = 82), and highly intercorrelated (*n* = 391), respectively. After the use of recursive feature elimination with logistic regression estimator and polynomial transformation, 92 features were finally selected. In the training set, the logistic regression classifier obtained an overall mean accuracy of 60%. In the test set, the accuracy of the ML classifier was 63%, with a sensitivity of 80%, specificity of 43%, and AUC of 66%. Conclusions: Radiomics and ML applied to pre-operative DCE-MRI in patients with IBC showed promises for the non-invasive prediction of ODXRS, aiding in selecting patients who will benefit from NAC.

## 1. Introduction

It is currently estimated that 1.6 million breast cancer (BC) cases occur worldwide each year, making it the most common solid neoplasm in women [1]. Substantial improvements in treatment over the past few decades have reduced overall breast cancer mortality, yet it remains a major cause of death [2]. BC is a highly heterogeneous disease, categorized into three groups based on different molecular profiles: luminal A/B (*ER*/*PgR*+, *HER2-* or +), *HER2*+ (*ER*/*PgR-*, *HER2*+) and triple negative (or basal-like, *ER*/*PgR*/*HER2-*) [3]. Therapeutical approaches are currently tailored according to each molecular subtype, including chemotherapy (either neoadjuvant or adjuvant) in *HER2*+ and TN subtypes. In hormone positive cancers, it is still debated whether cytotoxic chemotherapy could be beneficial [4]. Indeed, the risk of tumor recurrence has to first be assessed in such cases to fully address the cost/benefits ratio. This prognostic evaluation is made through multi-parameter, multi-analyte, and multi-gene tests, with Oncotype DX recurrence score (RS) being the most frequently used in clinical practice. This assay analyses a panel of 21 genes in women presenting early stage *ER*+, *HER2-* breast cancer, in order to determine a ‘Recurrence Score (RS)’ [5,6,7,8,9], (scored as 0–100), corresponding to the 10-year risk of recurrence, and is used to stratify low (<18), intermediate (18–30), or high (≥31) risk groups, predicting whether the patients may benefit from adjuvant chemotherapy or not [5,6,7,8]. As proved as valuable for the improvement of BC patients’ care, Oncotype DX RS has been increasingly used over the last years, and it is now included in all major international treatment guidelines [10,11]. Major limitations of Oncotype DX RS are represented by its high costs (maximum tariff €4487.02 [12]), which jeopardizes its widespread use, and by the fact that it is currently performed on histological specimen after surgical excision, so that patients can undergo chemotherapy only after surgical excision, in an adjuvant setting. The possibility to obtain a pre-operative Oncotype DX RS would allow the timely dentification of patients who could benefit from chemotherapy and, therefore, undergo it before surgical excision, in a neoadjuvant setting with all its related benefits, such as the in-vivo assessment of tumor response. In this light, recent advances in the field of diagnostic imaging and informatics, with the possibility to extract quantitative data not detectable by human eyes (radiomics) and build predictive models using artificial intelligence and machine learning (ML) techniques, could represent a powerful tool for the non-invasive prediction of Oncotype DX RS. Indeed, magnetic resonance imaging (MRI) is the modality of choice for pre-operative local tumor staging in BC, providing both morphological and functional data on tumor neoangiogenesis, expression of tumor aggressiveness [13]. A correlation between magnetic resonance imaging (MRI) features and Oncotype DX RS has been reported in literature, but investigations to explore the usefulness of radiomics features and artificial intelligence tools are currently lacking. Indeed, available ML and deep learning studies aimed to predict Oncotype DX RS using semantic features, but the role of radiomics features for this task has never been explored so far [14,15]. We, therefore, aimed to evaluate whether preoperative MRI-derived radiomics features employed by a ML algorithm could be used to non-invasively predict Oncotype DX RS in patients with *ER*+ and *HER2-* invasive BC.

## 2. Materials and Methods

### 2.1. Patient Population

A public dataset of breast MRI examinations was used, freely available at: https://wiki.cancerimagingarchive.net/plugins/servlet/mobile?contentId=70226903#content/view/70226903 accessed on 3 May 2022 [16]. The database includes 1150 consecutive breast cancer patients who underwent pre-operative breast MRI examinations from the 1 January 2000 to the 23 March 2014. Patients with no available Oncotype DX score were excluded. Oncotype DX score was considered as positive if greater than 16 in <50-year-old patients and greater than 26 in >50-year-old patients [17].

### 2.2. Imaging Data

Different scanners, from 1.5 to 3.0 T, were used for image acquisition, including: Avanto (Siemens Healthineers, Erlangen, Germany), Optima MR450w (GE Healthcare, Chicago, IL, USA), SIGNA EXCITE (GE Healthcare, USA), SIGNA HDx (GE Healthcare, USA), Signa HDxt (GE Healthcare, USA), Skyra (Siemens Healthineers, Erlangen, Germany), Trio (Siemens Healthineers, Erlangen, Germany), TrioTim (Siemens Healthineers, Erlangen, Germany). The following acquisition parameters were employed: TR from 3.54 to 6.74 ms, TE from 2.41 to 2.46 ms, slice thickness from 1.1 to 2.2 mm. The mean FOV was 12.345 cm. The following MRI sequences were available: a non-fat saturated T1-weighted sequence, a fat-saturated gradient echo T1-weighted pre-contrast sequence, and four post-contrast T1-weighted sequences acquired after the IV administration of contrast agent (a weight-based protocol of 0.2 mL/kg was performed). In our cohort, three types of contrast agents were used as it follows: gadobutrol (Gadavist, Bayer Healthcare, Berlin, Germany) in 2 (0.2%) patients, gadopentetate dimeglumine (Magnevist, Bayer Healthcare, Berlin, Germany) in 560 (60.8%) patients, and gadobenate dimeglumine (Multihance, Bracco, Milan, Italy) in 263 (28.5%) patients. Information on the employed contrast agent was not available for 97 (10.5%) patients. The median acquisition time between a pair of post-contrast sequences was 131 s.

### 2.3. Image Analysis

Handcrafted lesion segmentation was performed by a radiologist expert in breast imaging on first post-contrast images, as previously recommended [18] using a dedicated, freely available software (ITKSNAP, v3.8.0) [19] obtaining 3D regions of interest (ROIs) (Figure 1). In detail, tumor lesions were segmented using a semi-automated method avoiding the inclusion of macroscopic necrosis/cysts/neighboring vessels. A lower threshold was applied by the operator for the inclusion of the enhancing tumor tissue, exploiting its high contrast with the background parenchyma on subtracted images. If necessary, segmentation was manually adjusted to avoid the inclusion of surrounding non-tumor tissue.

Two senior radiology residents, with at least 2 years of experience in breast imaging, independently performed tumor segmentation on a sub-set of 30 patients randomly selected from the study cohort. This was done to assess features stability among segmentations performed by different operators through intraclass correlation coefficient (ICC) [20].

### 2.4. Radiomics Analysis

A dedicated open-source Python-based software (PyRadiomics, v3.0.1) was employed for image pre-processing and 3D radiomic feature extraction. Pixel resampling was performed with spacing set to 1 × 1 × 1 mm, to ensure rotational invariance of textural features [21,22]. Gray-level whole-image normalization was paired with scaling (=100) and voxel array shift (=300) with a resulting range of 0–600. A fixed bin width (=14) approach was used for grey level discretization. In regard to feature classes, 2D shape, first order, gray level co-occurrence matrix (GLCM), gray level run length matrix, gray level size zone matrix, gray level dependence, and neighboring gray tone difference matrix were extracted. All available features were calculated, except for GLCM sum average, as this proved to be redundant with other GLCM parameters. A two-way random effect, single rater, absolute agreement ICC was performed to assess radiomics features stability among segmentations performed by different operators [20,23]. Radiomics features with ICC ≥ 0.75 were considered stable and selected for the following steps. ICC analysis was conducted using the R “irr” package. The MinMax scaler with a 0–1 range was fitted on the training data alone and used to transform both training and test sets.

Thereafter, radiomics features were selected based on their variance between the two classes (positive and negative Oncotype DX RS). Low variant features (≤0.01) were thus excluded. Pairwise correlation matrix (Pearson’s r ≥ 0.8) was conducted to identify and remove radiomics features with high collinearity. Successively, the optimal number of radiomics features for the classification task was identified by a 10-fold cross-validated recursive feature elimination (RFECV) with a logistic regression (LBFGS solver) estimator.

Finally, a feature transformation was performed on the resulting feature set to generate a new feature matrix consisting of raw feature values and their interactions (i.e., second degree polynomial transformation set to produce interactions only) [24]. As done with scaling, the polynomial transformer was exclusively fit on the training data and then used to transform training and test sets. The pandas and scikit-learn Python packages were used for data processing. For the classification task, a logistic regression ML classifier algorithm was selected based on specific characteristics of the dataset (expected number of instances available, use of polynomial transformation, tabular nature of the data) and used with a 5-fold stratified cross-validation (scoring metric = balanced accuracy) during the random search tuning process. The search space was defined as follows: penalty = l1, l2 or elasticnet, solver: saga or LBFGS, l1 ratio = 0–1. This approach was chosen as expected to give a better estimation of generalizability [25], with the algorithm being trained in 4 data folds, and then tested in the remaining fold ensuring the class balance.

The Brier score and a calibration curve were calculated for the model on the test set, to assess prediction and calibration loss of predicted probability and lesion class.

The ML analysis was performed using the scikit-learn Python package. Accuracy metrics were computed with the same Python package and the caret R package [26].

## 3. Results

### 3.1. Patient Population

A total of 922 patients out of the 1150 were initially included in the dataset. Of these, patients with no available Oncotype DX score were further excluded, leaving a final population of 248 patients (mean age: 55.4; range 28.7–82.9) with 248 breast cancer lesions (Figure 2). Of these, 162 (65.3%) were in menopausal status. In patients younger than 50 years, Oncotype DX score was positive in 30 of 83 cases (36%). In patients older than 50 years, Oncotype DX score was positive in 57 of 165 cases (35%). Overall, 87 out of 248 (35%) patients had a positive Oncotype DX recurrence score. Of the 248 included patients, 166 (66%) were grouped in the training set, while the remaining 82 (33%) were grouped as test set. Histological features of included breast lesions are depicted in Table 1.

### 3.2. Feature Selection

A total of 1288 features were extracted. Of these, 517 were initially selected as stable at ICC analysis (ICC > 0.75). Of these, 82 resulted as non-variant between the two classes and 392 were highly intercorrelated. As a result, 44 features were retained and further analyzed by the RFECV with a LBFGS solver, which identified an optimal number of 13 features.

### 3.3. Machine Learning Analysis

In the polynomial logistic regression analysis, the data produced by the feature selection was transformed to a 92-feature set including raw data and radiomic feature interactions. In the training set, the classifier obtained an overall mean balanced accuracy of 60%. The best model configuration was characterized by an l2 penalty and LBFGS solver. In the test set, the accuracy of the trained ML classifier was 63%, with a sensitivity of 80% (95% CI: 0.66–0.89), specificity of 43% (95% CI: 0.29–0.59), positive likelihood ratio of 1.4 (95% CI: 1.03–1.94), negative likelihood ratio of 0.46 (95% CI: 0.23–0.92), and an area under the receiver operating characteristics curve (AUC) of 66% (Figure 3 and Figure 4). Accuracy metrics are summarized in Table 2.

## 4. Discussion

The aim of our study was to evaluate whether ML applied to DCE-MRI could be used to predict Oncotype DX RS in patients with *ER+* and *HER2-* invasive breast cancer. According to our findings, the ML classifier obtained an accuracy of 0.63% for the prediction of Oncotype DX score in the test set, showing a sensitivity of 80% and a ROCAUC of 0.66. Such data suggest a possible role of AI applied to pre-treatment DCE-MRI datasets for the non-invasive prediction of the risk of BC recurrence, with remarkable advantages in patients’ management. Oncotype DX RS has been used over the last ten years to stratify the need of a systemic treatment in BC patients, resulting in a decreased use of chemotherapy. Furthermore, a recent analysis by Schaafsma et al. showed that patients who underwent Oncotype DX RS tended to have a better survival compared to those who did not use the test [11]. Remarkably, high-risk patients undergoing chemotherapy showed an improved overall and BC-specific survival compared to patients with high-risk scores who did not receive chemotherapy. The same occurred to low-risk patients who forwent chemotherapy, showing a longer overall survival than patients with low-risk scores treated with chemotherapy. At present, Oncotype DX RS is recommended in patients with early *ER+* and *HER2-* BC who are candidates for surgical operation. The multigene essay is then performed on histological specimen to identify the risk of recurrence. According to test’s results, adjuvant chemotherapy is performed in high-risk patients. However, in such cases the in vivo assessment of the response is no longer feasible. Therefore, the preoperative stratification of recurrence risk could allow BC patients at high risk of tumor recurrence to perform neoadjuvant chemotherapy, allowing the in vivo assessment of the response and an easier, more conservative surgical approach. Previous attempts have been made to non-invasively and pre-operatively predict Oncotype DX score through AI techniques applied to diagnostic images. In 2017, Saha and colleagues retrospectively evaluated a comprehensive set of imaging features derived from DCE-MRI of 261 BC patients to predict the distant recurrence risk using Oncotype DX scores [14]. Two multivariate models were developed to discriminate between high and intermediate/low scores as well as between high/intermediate and low scores. As a result, the first model predicted high against low-intermediate Oncotype DX scores with an AUC of 0.77 (95% CI: 0.56–0.98), while low against high/intermediate score was predicted with a lower AUC of 0.51 (95% CI: 0.41–0.61). More recently, Kim et al. evaluated whether MRI could be used to predict Oncotype DX RS in patients with *ER+* and *HER2-* invasive breast cancer using a multivariate logistic regression analysis [15]. Semantic MRI features were used for this purpose, such as BI-RADS descriptors and kinetics features. As a result, round shape and low proportion of washout component were associated with low recurrence score, while heterogeneously dense, or scattered fibroglandular breast tissue, non-spiculated margins, and low proportion of persistent component, were associated with high recurrence score. Most of the available studies have been conducted using semantic and/or quantitative kinetic MRI features and a multivariate analysis [27]. So far, a ML analysis was carried out in 2020 by Jacobs et al., who used a new ML informatic system integrating clinical variables with multiparametric radiomics in 83 breast cancer lesions for the prediction of Oncotype DX score (low, intermediate, and high risk) [28]. An AUC of 0.89 was obtained in discriminating low from intermediate and high-risk groups. Conversely, a convolutional neural network was used by Ha et al. in 134 BC lesions, obtaining an overall accuracy of 84% in predicting patents with low Oncotype Dx RS compared to patients with intermediate/high Oncotype Dx RS [29]. Their results indicate the feasibility of utilizing the CNN algorithm to predict Oncotype Dx RS.

To the best of our knowledge, our study is the first investigation exploring the combination of radiomics features and machine learning for the prediction of a dichotomic Oncotype DX score, stratified according to patients’ age, as recently recommended [17]. Despite the relatively low diagnostic accuracy (AUC = 0.66), our findings further support the hypothesis that artificial intelligence techniques applied to diagnostic images may play a possible role for the non-invasive prediction of Oncotype DX score. This approach would allow clinicians to plan the treatment strategy more accurately, as it would indicate the need for neoadjuvant chemotherapy in high-risk cases. Among the available imaging techniques, MRI is the best candidate for this purpose, being the most suitable modality for local breast cancer staging and treatment response assessment after neoadjuvant chemotherapy [30,31]. Overall, this could result in a sensible gain in terms of time and costs, as Oncotype DX a more expensive and less available technique.

Limitations of our study are represented by its retrospective design and a relatively small sample size; furthermore, external testing is mandatory for the assessment of ML generalizability. Indeed, potential improvements of our model with a larger dataset, including its external validation using population from different institutions, may result in a useful predictive tool for determining patients’ likelihood of future breast cancer recurrences.

## 5. Conclusions

Based on our preliminary experience, a ML algorithm applied to DCE-MRI dataset proved to be feasible for the pre-operative, non-invasive prediction of Oncotype DX RS in BC. Our findings support the feasibility of radiomics and machine learning for the prediction of prognostic data in breast cancer, encouraging further, large-scale investigations to further improve the performance of the model and assess its generalizability.

## Figures and Tables

**Figure 1 cancers-15-01840-f001:**
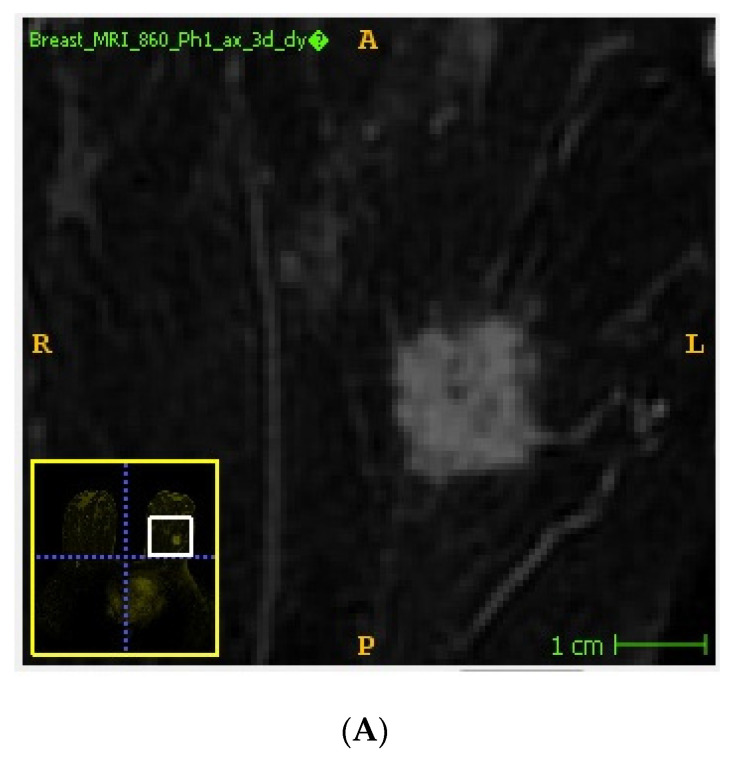
Example of tumor segmentation performed on subtracted DCE-MRI images using a semi-automated method. Unannotated (**A**) and annotated (**B**) DCE-MRI subtracted image showing an invasive ductal carcinoma of the left breast.

**Figure 2 cancers-15-01840-f002:**
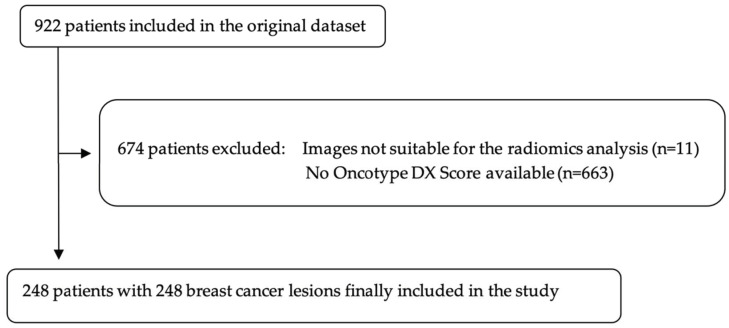
Flowchart of patients’ selection. Original dataset provided by Saha et al., 2018 [16].

**Figure 3 cancers-15-01840-f003:**
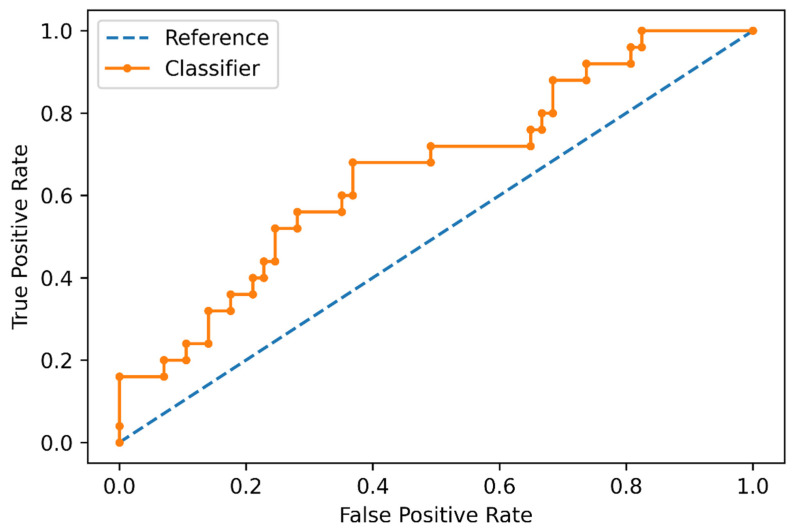
Receiver operating characteristic curve of the machine learning classifier for distinguishing negative and positive Oncotype DX Score in the test set.

**Figure 4 cancers-15-01840-f004:**
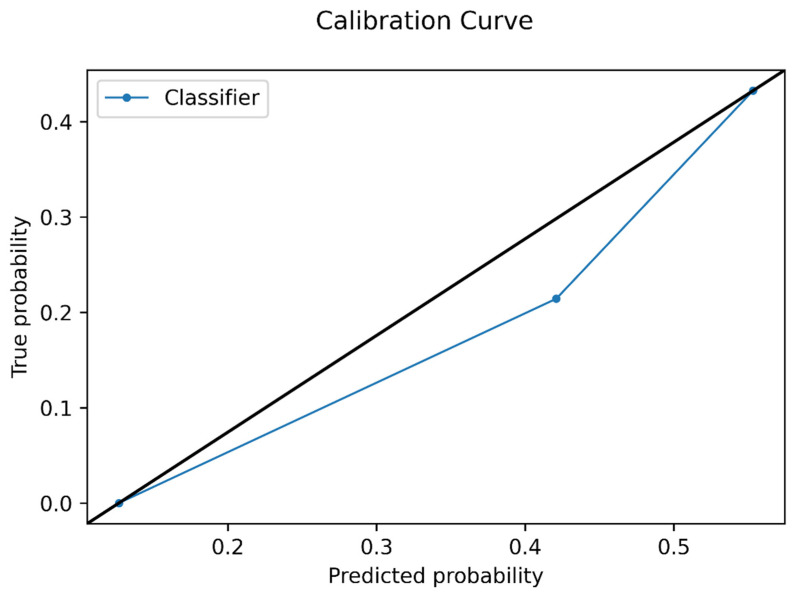
Calibration curve plot of the model in the test set. The average predicted probability is represented in the *x*-axis, while the proportion of positive Oncotype DX Score in reported in the *y*-axis.

**Table 1 cancers-15-01840-t001:** Histological features of included breast lesions.

	Negative Oncotype Score	Positive Oncotype Score	Total
***ER*+**	161 (64.9%)	87 (35.1%)	**248**
***PgR*+**	149 (67.1%)	75 (32.9%)	**222**
***HER2*-**	161 (64.9%)	87 (35.1%)	**248**
**Tumor Grade 1**	15 (55.5%)	12 (44.5%)	**27**
**Tumor Grade 2**	35 (68.6%)	16 (31.4%)	**51**
**Tumor Grade 3**	110 (65.5%)	58 (34.5%)	**168**
**Histologic type**			
**Lobular**	26 (86.6%)	4 (13.4%)	**30**
**Ductal**	96 (57.5%)	71 (42.5%)	**167**
**Mucinous**	2 (66.6%)	1 (33.4%)	**3**
**Not available**	37 (77.1%)	11 (22.9%)	**48**

**Table 2 cancers-15-01840-t002:** Accuracy metrics of the ML classifier in the test set.

Class	Precision	Recall	f1-Score	Total Cases
**Negative Oncotype DX score**	0.80	0.63	0.71	57
**Positive Oncotype DX score**	0.43	0.64	0.52	25

## Data Availability

The DCE-MRI dataset is available at: https://wiki.cancerimagingarchive.net/plugins/servlet/mobile?contentId=70226903#content/view/70226903, accessed on 3 May 2022. Further data supporting reported results can be provided by the corresponding author under reasonable request.

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
