# Peer review of "MRI Radiomics and Machine Learning for the Prediction of Oncotype Dx Recurrence Score in Invasive Breast Cancer"

_cancers, 2023, doi:10.3390/cancers15061840_

Round 1

Reviewer 1 Report

Thank you to the authors and editor for the opportunity to review this manuscript. 

The authors have employed a logistic regression model to classify patients pre-operatively into various risk categories based on Oncotype DX recurrence score, which would allow clinicians to assess risk non-invasively before a surgical specimen is extracted. 

Comments:

1. Consider the following sentence from the Abstract:  "Of these, 1244 were excluded as 771, 82 and 391 were excluded as 26 not stable (n=771), not variant (n=82) and highly intercorrelated (n=391)." You are missing the word "respectively" at the end of the sentence; else it's not easy to understand.

2. An AUC of 0.66 is not strong. May the authors add to the final paragraph of the Discussion to explain why the model did not perform better than this and what additional changes and strategies (besides sample size) might improve the results in the future? 

3. There appears to be a formatting issue on line 49 of p. 2

4. Line 55-57 of p. 2 is not clear: "Oncotype DX RS 55 would allow, when indicated, the performance of chemotherapy in the neoadjuvant set-56 ting, with a huge impact on the response assessment and treatment outcome." What would Oncotype DX RS allow in regards to the performance of chemotherapy?

5. The sentence on line 61-64 of p. 2 appears to be a run-on. 

6. Were features extracted only from the first post-contrast phase images? If so, can the authors explain why they limited analysis to this phase and whether they considered extracting features from kinetic maps (SER, PE, etc) as well or subtraction images?

7. The features did not seem to be corrected for batch effects (vendor, resolution, etc), as far as this reviewer understood from the Methods. This paper would be greatly strengthened if the authors considered a ComBat approach for batch correction.

8. Some more context in the Introduction would be useful, touching briefly on what approaches others have done to address non-invasive Oncotype DX RS prediction and why those approaches merit improvement, like in the Discussion.

9. Please explicitly state which covariates were included in your regression model. Did you use the radiomic features as separate covariates, features and clinical risk factors, or dimension-reduced features (PCA)?

10. Please motivate why you decided to split the data into 66.7/33.3% for training testing, respectively. Typically the splits are 80/20%. 

Author Response

Thank you to the authors and editor for the opportunity to review this manuscript. 

The authors have employed a logistic regression model to classify patients pre-operatively into various risk categories based on Oncotype DX recurrence score, which would allow clinicians to assess risk non-invasively before a surgical specimen is extracted. 

Comments:

1. Consider the following sentence from the Abstract:  "Of these, 1244 were excluded as 771, 82 and 391 were excluded as 26 not stable (n=771), not variant (n=82) and highly intercorrelated (n=391)." You are missing the word "respectively" at the end of the sentence; else it's not easy to understand.

The word “respectively” has been added at the end of the sentence, as suggested.

2. An AUC of 0.66 is not strong. May the authors add to the final paragraph of the Discussion to explain why the model did not perform better than this and what additional changes and strategies (besides sample size) might improve the results in the future? 

Considerations on the relatively low AUC and the need for our model to be improved through increased sample size and external validation using cases from different institutions have been added to the discussion section.

3. There appears to be a formatting issue on line 49 of p. 2

As missing in the original word file, we presume this formatting issue could be related to the new layout of the manuscript, and could be fixed later on during manuscript editing.

4. Line 55-57 of p. 2 is not clear: "Oncotype DX RS 55 would allow, when indicated, the performance of chemotherapy in the neoadjuvant set-56 ting, with a huge impact on the response assessment and treatment outcome." What would Oncotype DX RS allow in regards to the performance of chemotherapy?

This paragraph has been modified as follows (lines 84-90, p.2): Major limitations of Oncotype DX RS are represented by its high costs (maximum tariff € 4487.02 [12]), which jeopardizes its widespread use, and by the fact that it is currently performed on histological specimen after surgical excision, so that patients can undergo chemotherapy only after surgical excision, in an adjuvant setting. The possibility to obtain a pre-operative Oncotype DX RS would allow a timely identification of patients who could benefit from chemotherapy and therefore undergo it before surgical excision, in a neoadjuvant setting with all its related benefits, such as the in-vivo assessment of tumor response.

5. The sentence on line 61-64 of p. 2 appears to be a run-on. 

The sentence has been modified as follows (lines 95-98, p.2): “Indeed, magnetic resonance imaging (MRI) is the modality of choice for pre-operative local tumor staging in BC, providing both morphological and functional data on tumor neoangiogenesis, expression of tumor aggressiveness [13]”

6. Were features extracted only from the first post-contrast phase images? If so, can the authors explain why they limited analysis to this phase and whether they considered extracting features from kinetic maps (SER, PE, etc) as well or subtraction images?

According to breast cancer lesions’ kinetics, early time points are more representative of tumor aggressiveness, related to its neoagiogenesis. Conversely, in later time points, enhancement due to inflammatory changes and lower aggressive tumor portions also occurs. We therefore aimed to use the first post-contrast phase in which the lesion was clearly visible to better depict those biological properties that might be on the basis of tumor recurrence. We specifically aimed to assess the contribution of radiomics features for this purpose.

7. The features did not seem to be corrected for batch effects (vendor, resolution, etc), as far as this reviewer understood from the Methods. This paper would be greatly strengthened if the authors considered a ComBat approach for batch correction.

Batch effect correction is a complex and somewhat debated method [10.1186/s12859-020-03559-6, 10.1093/biostatistics/kxv027], and has the limitation of requiring joint analysis of training and test data, potentially inducing information leakage. This approach also reflects poorly clinical practice, as it would not be feasible to apply ComBat to new cases in a real-world setting.

8. Some more context in the Introduction would be useful, touching briefly on what approaches others have done to address non-invasive Oncotype DX RS prediction and why those approaches merit improvement, like in the Discussion.

A brief mention to previous studies addressing non-invasive Oncotype DX RS has been added to the introduction section, as suggested (lines 100-102, p.2).

9. Please explicitly state which covariates were included in your regression model. Did you use the radiomic features as separate covariates, features and clinical risk factors, or dimension-reduced features (PCA)?

The polynomial regression was entirely based on the radiomics features obtained from the feature selection process.

10. Please motivate why you decided to split the data into 66.7/33.3% for training testing, respectively. Typically the splits are 80/20%.

There is no hard rule on the appropriate proportion to split the construction (training + validation) and hold out test set. Typically, 33% or 20% are commonly employed, as also 25% (but not only, see for example this discussion: https://stats.stackexchange.com/questions/237691/any-theory-on-how-to-split-the-data). In this study, 33% gave us a decent sample size in the test set, without sacrificing too much in terms of construction set size. Obviously, proportion of the split was not an hyperparameter object of tuning but chosen a priori.

Reviewer 2 Report

General comments:

1. In ML it is common to use three datasets: training, validation and test - compare, e.g. https://doi.org/10.1186/s13040-017-0155-3. The training dataset is used to train the weights of the algorithm, while the validation DS is a method to determine hyperparameters. Why did the authors deviate from this scheme and what are the implications of doing so?

2. The motivation for the study is to avoid expensive Oncotype DX (post-operative) by replacing it with non-invasive MRI. Could the authors give numbers for the cost of both procedures?

3. I could not find any information on the layers/interconnections of the ML model employed. In order for this research to be repeatable this information must be provided.

4. Could one not constrain some variables so thet the L-BFGS-B algorithm also availabe in scikit might be a better choice than L-BFGS alone?

5. The achieved accuracy of 63% is an intermediate range. In the discussion, could the authors add some advice on how to possibly enhance the accuracy?

Minor comments:

Abstract lines 19-20: I cannot follow the definition of positive ODXRS with the confusing use of ">" and "<" signs. Please clarify.

Abstract line 32: Missing "%" sign

Please make sure that all abbreviations are introduced.

Line 51. Remove trailing "." after references for the cmplete sentence before

Lines 64-65: Please give a reference where the correlation between MRI and Oncotype DX RS is described.

Lines 104-105: Please detail which semi-automated segmentation was used.

Section 2.4: Could you also include references to the recommendations of the software developers mentioned in mutliple positions here?

Author Response

1. In ML it is common to use three datasets: training, validation and test - compare, e.g. https://doi.org/10.1186/s13040-017-0155-3. The training dataset is used to train the weights of the algorithm, while the validation DS is a method to determine hyperparameters. Why did the authors deviate from this scheme and what are the implications of doing so?

We agree with the reviewer, and in fact our approach did follow the typical scheme. Simply, the validation of the model (for tuning purposes) was performed not on a single validation dataset but through 5-fold CV. This allows for a more robust estimate of the model’s general performance. Then, the final model was used to perform inferences on the hold out test set to obtain an unbiased estimate of its accuracy.

2. The motivation for the study is to avoid expensive Oncotype DX (post-operative) by replacing it with non-invasive MRI. Could the authors give numbers for the cost of both procedures?

The maximum tariff of Oncotype DX score has been added in the Introduction section (line 84, p2).

3. I could not find any information on the layers/interconnections of the ML model employed. In order for this research to be repeatable this information must be provided.

We apologize for the misunderstanding. In this paper, we did not use a DL model but a polynomial regression, which does not have these hyperparameters/characteristics. The hyperparameter search space for the model tuning is specified in the methods. For your convenience: “[…] penalty = l1, l2 or elasticnet, solver: saga or lbfgs, l1 ratio = 0-1”.

4. Could one not constrain some variables so thet the L-BFGS-B algorithm also available in scikit might be a better choice than L-BFGS alone?

We apologize, but were not able to find a scikit-learn implementation of the lbfgs-b solver in the scikit-learn logistic regression classifier (https://scikit-learn.org/stable/modules/generated/sklearn.linear_model.LogisticRegression.html). This solver is available in scipy, but this was not the Python package employed in the analysis.

5. The achieved accuracy of 63% is an intermediate range. In the discussion, could the authors add some advice on how to possibly enhance the accuracy?

As also pointed out by Reviewer 1, considerations on possible improvements of the machine learning model have been added in the discussion section.

Minor comments:

Abstract lines 19-20: I cannot follow the definition of positive ODXRS with the confusing use of ">" and "<" signs. Please clarify.

We apologize for the lack of clarity and modified this sentence as follows: ODXRS was obtained on histological tumor samples and considered as positive if greater than 16 and 26 in under and over 50 year-old patients, respectively.

Abstract line 32: Missing "%" sign

We thank the Reviewer for having noticed this typo, “%” has been now added.

Please make sure that all abbreviations are introduced.

All abbreviations have been checked and amended.

Line 51. Remove trailing "." after references for the complete sentence before

Reference style has been amended.

Lines 64-65: Please give a reference where the correlation between MRI and Oncotype DX RS is described.

References have been added with a brief mention to previous studies exploring the correlation between MRI and Oncotype DX RS (lines 100-102, p.2).

Lines 104-105: Please detail which semi-automated segmentation was used.

A lower threshold was applied by the operator for the inclusion of the enhancing tumor tissue, exploiting its high contrast with the background parenchyma on subtracted images. If necessary, segmentation was manually adjusted to avoid the inclusion of surrounding non-tumor tissue. These details have been added in the revised version of the manuscript.

Section 2.4: Could you also include references to the recommendations of the software developers mentioned in multiple positions here?

The only reference missing regarding recommendations is in relation to the choice of the logistic regression model. In this case, we paired the scikit-learn cheat sheet (https://scikit-learn.org/stable/tutorial/machine_learning_map/index.html) and well-established knowledge on the value of simple models in relatively small datasets (https://www.kaggle.com/code/rafjaa/dealing-with-very-small-datasets).